# Magnesium Oxide Nanoparticles for the Adsorption of Pentavalent Arsenic from Water: Effects of Calcination

**DOI:** 10.3390/membranes13050475

**Published:** 2023-04-28

**Authors:** Shaymala Mehanathan, Juhana Jaafar, Atikah Mohd Nasir, Ahmad Fauzi Ismail, Takeshi Matsuura, Mohd Hafiz Dzarfan Othman, Mukhlis A. Rahman, Norhaniza Yusof

**Affiliations:** 1Advanced Membrane Technology Research Centre (AMTEC), Faculty of Chemical and Energy Engineering, Universiti Teknologi Malaysia, Johor Bahru 81310, Malaysia; 2Centre for Diagnostic, Therapeutic and Investigative Studies (CODTIS), Faculty of Health Sciences, Universiti Kebangsaan Malaysia, Bangi 43600, Malaysia; 3Department of Chemical and Biological Engineering, University of Ottawa, Ottawa, ON K1N 6N5, Canada

**Keywords:** adsorption, calcination, pentavalent arsenic, water treatment, pore size distribution

## Abstract

The occurrence of heavy metal ions in water is intractable, and it has currently become a serious environmental issue to deal with. The effects of calcining magnesium oxide at 650 °C and the impacts on the adsorption of pentavalent arsenic from water are reported in this paper. The pore nature of a material has a direct impact on its ability to function as an adsorbent for its respective pollutant. Calcining magnesium oxide is not only beneficial in enhancing its purity but has also been proven to increase the pore size distribution. Magnesium oxide, as an exceptionally important inorganic material, has been widely studied in view of its unique surface properties, but the correlation between its surface structure and physicochemical performance is still scarce. In this paper, magnesium oxide nanoparticles calcined at 650 °C are assessed to remove the negatively charged arsenate ions from an aqueous solution. The increased pore size distribution was able to give an experimental maximum adsorption capacity of 115.27 mg/g with an adsorbent dosage of 0.5 g/L. Non-linear kinetics and isotherm models were studied to identify the adsorption process of ions onto the calcined nanoparticles. From the adsorption kinetics study, the non-linear pseudo-first order showed an effective adsorption mechanism, and the most suitable adsorption isotherm was the non-linear Freundlich isotherm. The resulting *R*^2^ values of other kinetic models, namely Webber-Morris and Elovich, were still below those of the non-linear pseudo-first-order model. The regeneration of magnesium oxide in the adsorption of negatively charged ions was determined by making comparisons between fresh and recycled adsorbent that has been treated with a 1 M NaOH solution.

## 1. Introduction

The scope of arsenic contamination in drinking water and the threat it poses to global health are much more widespread than previously believed. As many as 140 million people worldwide may have been exposed to drinking water with arsenic contamination levels higher than the World Health Organization’s (WHO) provisional guideline of 10 μg/L [1]. The uses of arsenic in medicine [2], science, and technology [3] are being overshadowed by its behavior as a homicide in our daily lives. Arsenic is mobilized by natural weathering reactions, biological activity, geochemical reactions, volcanic emissions, and other anthropogenic activities. Soil erosion and leaching contribute to arsenic in the oceans in dissolved and suspended forms [4]. Most environmental arsenic problems are the consequence of mobilization under natural conditions. However, mining activities, the combustion of fossil fuels, the use of arsenic pesticides, herbicides, and crop desiccants, and the use of arsenic additives in livestock feed create additional impacts. Furthermore, due to the unique characteristics of arsenic doping chemistry, there are no replacement elements for arsenic in the semiconductor industry. Arsenic is one of the critical elements used in the manufacturing of silicon-based semiconductors [5]. Arsenic exists in the −3, 0, +3, and +5 oxidation states [6]. Two forms are common in natural waters: arsenite (AsO_3_^3−^) and arsenate (AsO_4_^3−^), referred to as arsenic(III) and arsenic(V). Pentavalent (+5) or arsenate species are AsO_4_^3−^, HAsO_4_^2−^, and H_2_AsO_4_^−^ while trivalent (+3) arsenites include As(OH)_3_, As(OH)_4_^−^, AsO_2_OH_2_^−^, and AsO_3_^3−^. Pentavalent species predominate and are stable in oxygen-rich aerobic environments. Trivalent arsenites predominate in moderately reducing anaerobic environments such as groundwater [7]. As(III) is the second most stable form and is said to be more toxic and mobile than As(V) [8]. As(III) is needed to be converted into As(V) by chemical or biological methods to reduce its toxicity and the difficulty of subsequent treatment [9,10,11].

Various arsenic-removing technologies include chemical precipitation, ion exchange, chemical oxidation, reduction, reverse osmosis, ultrafiltration, electrodialysis, and adsorption [12]. Among these, adsorption is a prominent process being executed when it comes to removing arsenic from water resources. This is because adsorption is the most efficient, as the other techniques possess limitations such as the generation of a large amount of sludge, low efficiency, sensitive operating conditions, and costly disposal. The adsorption method is emerging as a potentially preferred alternative for the removal of heavy metals, especially arsenic, because it provides flexibility in designing adsorbents, high-quality treated effluent, is reversible, and the adsorbent can be regenerated [13].

There are plenty of natural and synthesized adsorbents being reported for arsenic adsorption from water. H. Genç et al. [14] reported seawater-neutralized red mud (Bauxsol), a waste from aluminum manufacturing, as an adsorbent for removing As(V) (arsenate) from water. The author described this batch study as a cost-efficient pre-treatment method with some suggestions to further modify the natural adsorbent. Many solid materials have been used as adsorbents, but nanomaterials have been reported to be more effective. However, some of the nanomaterials are toxic, difficult to regenerate after adsorption, and ineffective in the presence of water constraints. Oxides and hydroxides of iron, aluminum, manganese, and magnesium are able to prevent these shortcomings as they are environmentally friendly and have been used to remove a variety of water pollutants. These oxides adsorb anions as well as cations effectively from their aqueous solutions. Arsenate ions can bind to the metal oxide surface via chemical or physical adsorption. High surface area-to-volume ratio, high-level surface defects, high density of reactive sites on the surface, and high intrinsic reactivity of surface sites are some of the characteristics of metal oxide that provide better sorption of arsenic. The metal (As) and ligand (O) characteristics of oxyanions of arsenic provide the easiest way of approaching the metal oxide surfaces via surface complexation or ligand exchange, which leads to the formation of mono- or bi-dentate complexes [15]. Therefore, metal oxides have a large potential for arsenic remediation from water. In general, metal oxides exhibit a higher removal capacity for As(V) [16].

In that case, due to its efficiency and cost effectiveness, magnesium oxide is being used in the wastewater treatment industry to filter out suspended solids and precipitate dissolved heavy metals [17,18]. Since the pH of the zero point of charge (pH_zpc_) is 12.4, magnesium oxide has been utilized as an anion adsorbent because of the favorable electrostatic attraction provided by the Mg atom [19]. As an example, in previous studies, Mg-bearing minerals and materials were used for the adsorption of arsenic [20,21,22]. Apart from that, magnesium oxide could be able to adsorb phosphorus with minimal environmental impact and harmful by-product generation due to the small ionic radius and high charge density of magnesium atoms. Magnesium oxide nanoparticles are cost-friendly, exhibit a sufficient number of surface reactive sites and isolated hydroxyl groups, and exhibit a high affinity for the adsorption of negatively charged phosphate ions [23,24]. Magnesium oxide is the most promising anions for the removal of pollutant ions like fluoride (F^−^) and borate (B(OH)_4_^−^) [25,26].

Most magnesium oxide-related adsorption studies are related to the calcination process due to the temperature-dependent structure of the material. The calcination temperature is said to have the main role in determining the surface structure and physicochemical properties of this material, which are responsible for its overall performance. Along with the surface change of magnesium oxide from a smooth appearance to a structure composed of nano-sized grains, the crystal structure has also evolved from meso-crystal to polycrystal, then to pseudomorph, and finally to cubic single crystal with the increase of calcination temperature in the range of 400–1000 °C [27].

The calcination process can have a high impact on the adsorption process, as it plays a vital role in transforming the physical nature of the adsorbent. In other words, calcination is usually attempted to improve the surface properties of the adsorbent [28]. Generally, high-temperature calcination processes have been shown to enhance the adsorbent’s surface capacity and deform surface textural and mineralogical properties [29]. The process of calcination has been reported as one of the effective methods that can help increase adsorbent hardness and decrease its water adhesion, preventing the breakage of the adsorbent. Four types of magnesium oxides can be produced by calcining their precursors [30]: light-burned or caustic-calcined magnesium oxides (calcined at 700–1000 °C), with the highest reactivity and greatest specific surface area; hard-burned magnesium oxides (calcined at 1000–1500 °C), with lower reactivity and specific surface area than those of light-burned magnesium oxides; dead-burned magnesium oxides or periclase (calcined at 1400–2000 °C), with the lowest specific surface area, making them almost unreactive; fused magnesium oxides (calcined at 2800 °C) with the lowest reactivity.

A study optimizing the preparation of pure magnesium oxides stated that the highest reactivity of the material was achieved when a basic magnesium carbonate was calcined at a temperature of 666.99 °C [31]. In a study of As(V) adsorption by mesoporous aluminum magnesium oxide composites, it was reported that the calcination temperature of 400 °C for these composites has provided highly ordered mesopores with a high surface area and pore volume, which has further provided an extremely high adsorptive capacity [21]. In another study by Mahmood et al., a decrease in arsenic adsorption efficiency by mixed oxide was due to increased calcination temperature, which decreased the surface area of the adsorbent [32]. All of the mentioned efforts were focused on thermally decomposing magnesium oxide precursors such as magnesium hydroxides, magnesium carbonates, magnesium chlorides, magnesium nitrate, magnesium acetates, and composites [33]. Each and every precursor has its own favorable calcination temperature to make productive magnesium oxide material with respect to its required performances. Equations (1) and (2) show the thermal decomposition of two common precursors of magnesium, magnesium hydroxide (Mg(OH)_2_) and magnesium carbonate (MgCO_3_), to produce magnesium oxide (MgO) [34].
(1)$$\mathrm{Mg}{\left(\mathrm{OH}\right)}_{2\;}\to \mathrm{MgO}+H_{2}O$$
(2)$${\mathrm{MgCO}}_{3}\to \mathrm{MgO}+{\mathrm{CO}}_{2}$$

At low temperatures of calcination, the loss of gases takes place, leaving a very porous structure with a large internal surface area and great reactivity for adsorption [35]. The increase in surface area when heating the precursors is due to the increase in crystallite structure, and the material has more components like carbonates, oxalates, and acetates to get heated, causing denser voids to form for the preparation of pure material. On the other hand, further calcining readily available magnesium oxides usually causes the surface area to decrease and an increase in the Mg^2+^ amount in the material. The influence of calcination environments (air or nitrogen) and temperatures (lower or higher) on adsorbent composites for As(V) and As(III) removal has been rarely discussed in the literature [36].

From other perspectives, the platform for heavy metal removal has become more convenient, easier, and effective with the use of adsorptive membranes. Adsorptive membranes are stable and able to prevent fouling issues. The flexibility and simplicity of fabricating long-lasting adsorptive membranes have opened the door for the discovery of more efficient adsorbents for water treatment applications, specifically heavy metal removal. For an effective adsorptive membrane to emerge, the characteristics of an adsorbent play an important role. Adsorbents with a higher pore volume and more active sites are able to create highly adsorptive membranes. Providing a higher adsorptive capacity is a non-compromised role that needs to be played by an adsorbent because a slight drop in the adsorptive capacity is usually observed during the transformation of adsorbents into adsorptive membranes due to less exposure of the adsorbent to pollutants. In that case, magnesium oxides have taken many forms in previous studies to remove a variety of metal ions such as Zn^2+^, Cd^2+^, Cu^2+^, and Cr^3+^ [37].

In the current work, we evaluated the adsorption performance of commercial magnesium oxide nanoparticles specifically calcined at a temperature of 650 °C for the removal of As(V) from aqueous solutions. The surface texture of the uncalcined and calcined nanoparticles is compared, and mineralogical variations are studied using FTIR, BET, XRD, and SEM techniques. The present study is an attempt to increase the electrostatic attraction of magnesium oxide to enable the capture of arsenate ions, to propose the capability of calcining at 650 °C, and to investigate the adsorption capacity of these nanoparticles for the elimination of As(V), which may help to elucidate the influence of solution pH, As(V) concentration, contact time, and the dosage effect of adsorbents. Table 1 shows the comparisons of other magnesium attached adsorbents related adsorptive studies.

## 2. Materials and Methods

### 2.1. Materials

Magnesium oxide nanoparticles from Nanjing XFNANO Materials Tech Co., Ltd. (Nanjing, China) and sodium dibasic arsenate heptahydrate purchased from Sigma Aldrich (St. Louis, MO, USA). The water utilized for the examinations was sanitized with a water decontamination framework Milli-Q water (resistivity 18.2 MΩ·cm, Merck Millipore, Lyon, France). All chemicals were analytical grade and were used without further purification.

### 2.2. Calcination of Magnesium Oxide Nanoparticles

Magnesium oxide nanoparticles were calcined at a temperature of 650 °C at a heating rate of 5 °C/min for 1 h and stored in dry bottles.

### 2.3. Characterization of MgO-650 °C Adsorbents

#### 2.3.1. Morphology Analysis

A high-resolution scanning electron microscope (SEM) (Brand: JEOL, Tokyo, Japan, Model: JSM-IT300LV) equipped with EDX software (Brand: Oxford, Abingdon, UK, Model: X-max 50) was employed to capture the morphology, structure, and distribution of metal elements within uncalcined and calcined magnesium oxide nanoparticles.

#### 2.3.2. Measurement of Total Active Surface Area and Crystallite Size

The total surface area and pore volume of uncalcined and calcined magnesium oxides were evaluated using nitrogen adsorption-desorption analysis. The multipoint measurement was conducted using Micromeritics ASAP 2010 and analyzed at 77 K. The sample was degassed at 130 °C for 1 h under a helium blanket and then directly placed in the holder for analysis. The total surface area of the nanoparticles was calculated using the Brunauer-Emmet-Teller (BET) equation. While the total pore volume was measured by converting the adsorption amount at P/P_0_ = 0.95 to a volume of liquid adsorbate. The structure and crystallite size of magnesium oxide nanoparticles were confirmed by powdered X-ray diffraction (XRD, Rigaku, Tokyo, Japan) in the range of 2-theta from 20° to 90° and operated at 40 kV and 30 mA.

#### 2.3.3. Detection of Functional Groups

Fourier transform infrared was used to detect the functional group occurring within the nanoparticles before and after the adsorption of arsenic, respectively. Before the analysis, the calcined magnesium oxide nanoparticles were shaken for 7 h in 50 mL of 10 mg/L arsenate solution, then dried at 100 °C in the oven for two days to remove any excess water.

#### 2.3.4. Zeta Potential Measurement

The zeta potential of calcined magnesium oxide particles was detected by the concept of laser diffraction-based dynamic light scattering and operated by Zetasizer Nano (Malvern Instruments, Malvern, UK). The zeta potential of the particle was calculated by determining the electrophoretic mobility and then applying the Henry equation as shown in (3):(3)$$U_{E}=\frac{2\varepsilon \zeta f\left(k\alpha \right)}{3\eta }$$
where, *U_E_* is electrophoretic mobility; *ζ* is zeta potential; *ε* is a dielectric constant; *η* is viscosity; and *ƒ*(*Ka*) is Henry’s function. The electrophoretic mobility was obtained by performing an electrophoresis experiment on the sample and measuring the velocity of the particles using laser doppler velocimetry (LDV). PZC of calcined magnesium oxide nanoparticles was measured by varying the pH (1.0 to 12.0) of the titrants in the system. The pH of the titrants was adjusted by 0.25 M HCl and 0.25 M NaOH. Zeta potential readings could determine the surface charge of the adsorbent.

### 2.4. Batch Adsorption Experiments

#### 2.4.1. Preparation of Arsenate Stock Solution

A stock solution of arsenate (1000 mg/L) was prepared by dissolving 2.246 g of sodium arsenate dibasic heptahydrate, Na_2_HAsO_4_·7H_2_O, in 1 L of deionized water. The 1000 mg/L of arsenate solution is diluted to 100 mg/L by adding 10 mL of the stock solution to 90 mL of deionized water, followed by the neutralization of 10 mL of 37.5% HCl and 10 mL of 3.6 M KOH. The stock solution prepared cannot be used for more than 2 days after preparation due to the high volatility and instability of arsenate ions. The stock solution was kept in the refrigerator at a temperature below 4 °C to inhibit microorganism formation.

#### 2.4.2. Measurement of Arsenate Concentration

An atomic absorption spectrophotometer (AAS), the Pinaacle 900T, from PerkinElmer (Waltham, MA, USA), was used to measure total arsenic concentration. An element lamp called an electrode discharge lamp (EDL) was externally installed in the instrument to measure arsenic concentration. The sample solutions were analyzed by direct aspiration into an air-acetylene flame, and absorbance was measured at the wavelengths of 188.980 nm and 193.696 nm. The sample solutions were aspirated by the nebulizer; the volume of solutions required is 2.0 mL. Flow rates of acetylene and air were adjusted to 2.5 L/min and 8 L/min, respectively, as specified by the manufacturer to give optimum sensitivity for arsenic element measurement. Calibration curves were plotted for each day of analysis by using five standard solutions and one blank solution. The absorbance values of the standard and sample solutions were determined by an average of replicate measurements every five seconds. The absorbance values of the sample solutions were determined from the calibration curves.

#### 2.4.3. Kinetics

Kinetic studies of arsenate adsorption on calcined magnesium oxide adsorbent were executed by placing 0.05 g of the particles into a 250 mL glass-covered conical flask containing 100 mL of arsenate solution at three different initial concentrations of arsenate solution, which were 10 mg/L, 20 mg/L, and 50 mg/L. The pH of the arsenate solution was fixed at pH 7.0 ± 0.1, which represents the standard pH of drinking water. The conical flasks were then shaken by a rotary shaker at a rate of 200 rpm. The contact time of adsorption was fixed at 7 h. Within the period, at different time intervals, 10 mL aliquots were derived from the suspension of arsenate solution for arsenate detection by AAS. The kinetic studies were performed at room temperature, 25 ± 2 °C.

#### 2.4.4. Isotherm

The adsorption isotherm analyses of arsenate adsorption by calcined magnesium oxide adsorbent were accomplished. By employing a dosage of adsorbent of 0.50 g/L, the experiment was performed at room temperature, 25 ± 2 °C, with a rotation of 200 rpm. The pH of the solutions was adjusted to pH 7.0 ± 0.1. The initial and residual concentrations of arsenate solution were detected by AAS after 7 h.

#### 2.4.5. PH Effect

Since the removal of arsenate is mainly governed by the calcined particles, the pH effect of the adsorbent was analyzed. The pH effect study was conducted by preparing arsenate solutions at various pHs ranging from 2.0 ± 0.1 to 12.0 ± 0.1. The initial concentration of arsenate solutions was fixed at 10 mg/L. The calcined magnesium oxide adsorbent (0.05 g) was shaken (200 rpm) in 100 mL of 10 mg/L arsenate solutions for 7 h due to the equilibrium time reported in the kinetic study. After 7 h, the equilibrium concentration of arsenate was measured using a graphite furnace equipped with an atomic absorption spectrophotometer (AAS) from Perkin-Elmer (Waltham, MA, USA).

#### 2.4.6. Adsorbent Dosage Effect

The adsorbent dosage effect on arsenate adsorption was tested by further increasing the amount of calcined magnesium oxide adsorbent. The adsorbent dosages of 0.5, 1.0, 2.0, 3.0, 4.0, and 5.0 g/L were attempted at the initial concentration of arsenate solution fixed at 10 mg/L. After 7 h, the equilibrium concentration of arsenate was measured using AAS.

#### 2.4.7. Data Analysis

After completing the adsorption process, the equilibrium arsenate adsorption capacity, $Q_{e}$ (mg/g) and the adsorption efficiency,$\;A_{E}$ (%) were calculated with Equations (4) and (5):(4)$$Q_{e}=\frac{\left(C_{o}-C_{e}\right)V}{m}$$
(5)$$A_{E}=\frac{\left(C_{O}-C_{e}\right)}{C_{O}}\times \text{100\%}$$
where, $C_{o}$, $C_{e}$, *V*, and *m* indicate the initial and equilibrium concentrations of arsenate (mg/L), the volume of arsenate solution (L), and the weight of the adsorbent used (g), respectively.

For kinetic adsorption, the experimental data were suited to pseudo-first order and pseudo-second-order models. The pseudo-first-order model is shown in non-linear form as Equation (6):(6)$$Q_{t}=Q_{e}\;\left[1-\exp\left(-K_{1}t\right)\right]$$
where $K_{1}$ (1/min) is defined as a pseudo-first-order adsorption rate constant. $Q_{e}$ (mg/g) and $Q_{t}$ (mg/g) are the capacities for arsenate adsorption at equilibrium and at their respective contact times, respectively.

A pseudo-second-order model in non-linear form is expressed by Equation (7):(7)$$Q_{t}=\left[\frac{t}{1/\left(K_{2}\;\cdot \;Q_{e}^{2}\right)}+\left(\frac{t}{Q_{e}}\right)\right]$$
where $K_{2}$ (g/(mg·min)) is the pseudo-second-order adsorption rate constant.

For isotherm analysis, the experimental data were fitted into the Langmuir and Freundlich isotherm models. The non-linear form of the Langmuir isotherm equation is designated as Equation (8):(8)$$Q_{e}=\frac{Q_{m}\;\cdot \;K_{L\;}\cdot \;C_{e}}{1+\left(K_{L}\;\cdot \;C_{e}\right)}$$
where $C_{e}$ (mg/g) is specified as the concentration of arsenate at equilibrium, $Q_{e}$ (mg/g) is specified as the equilibrium adsorption capacity of the adsorbent, while $Q_{max}\;$ is specified as the maximum adsorption capacity of the adsorbent, and $K_{L}$ (L/mg) is the Langmuir adsorption constant. The non-linear form of the Freundlich isotherm is shown as Equation (9):(9)$$Q_{e}=K_{F}\cdot C_{e}^{1/n}$$
where $K_{F}$ and *n* are Freundlich constants related to the adsorption capability and degree of adsorbent saturation.

#### 2.4.8. Regeneration Study of Calcined Magnesium Oxide Adsorbents

Two cycles of adsorption and desorption were carried out to evaluate the reusability of the prepared magnesium oxides. For the adsorption test, an adsorbent dosage of 0.05 g/L was added to an arsenate solution of 10 mg/L and stirred for 7 h at a fixed pH of 7.0. Then, the adsorbent was separated and collected from the solution using vacuum filtration. The residual arsenate concentration in the filtrate was measured using the AAS. For the desorption test, the arsenate-adsorbed adsorbent was added to a 100 mL 1 M NaOH solution. The mixture was stirred for 6 h, and later, the regenerated adsorbent was separated from the NaOH solution. After washing and drying, it was used in the next adsorption-desorption cycle. The steps were repeated for four cycles. The reusability of the adsorbents was evaluated based on the two-cycle regeneration experiment. The reusability of the calcined magnesium oxides was analyzed by measuring regeneration efficiency, which was calculated using Equation (10):(10)$$Regeneration\;efficiency\;=\frac{\begin{array}{c} Concentration\;of\;As\left(V\right)\;\\ adsorbed\left(recycled\right) \end{array}}{\begin{array}{c} Concentration\;of\;As\left(V\right)\\ adsorbed\left(fresh\right) \end{array}}\times 100\%$$

## 3. Results

### 3.1. Characterization of Calcined Magnesium Oxide Nanoparticles in Comparison with the Uncalcined

#### 3.1.1. Surface Morphology and Textural Behavior

Figure 1 shows the morphology of magnesium oxide, both calcined and uncalcined. Figure 1a,b is the overall view of the particles, while the others are views of one grain of a nanoparticle under certain magnifications, respectively. For uncalcined particles, as in Figure 1a,c,e, it demonstrates a fine and smooth structure. After treating the nanoparticles at 650 °C, some fractures appeared on the surface of the particles, as shown in Figure 1b,d,f, which could be attributed to the consequence of the release of crystal water and impurities. Magnesium oxide that has been formed by heating the precursors is used again for the arsenate adsorption purpose in this study. The heated structure is being introduced back into the heating process, which generates porous characteristics. During calcination, if the heating rate is too fast, the escape of CO_2_ and H_2_O will easily lead to the destruction of the laminate structure [44]. Furthermore, as shown in the figure, the crystals are too tiny to be observed without calcination effects. When heated up to 650 °C, the grains become much denser and observable. The process of heating the ore to a high temperature below its melting point to bring about thermal decomposition could make the ore porous. This is because during calcination, the volatile impurities and volatile oxides are expelled from the ore. At low calcination temperatures, the porous aggregates that comprise the particles are sufficiently strong to resist breakdown during compaction, so that the density before firing is low, and, in addition, the very high sintering activity of the small crystallites leads to an internal shrinkage within the aggregate that can open up interaggregate pores [45].

The elemental composition of both samples is shown in Figure 2. The composition of magnesium (Mg) atoms in the calcined sample is greater than in the uncalcined sample. With a calcination temperature of 650 °C, decomposition has taken place, which has caused the content of the oxygen (O) atom to decrease. This is because in the calcination process, the O atom will become gas and release from the surface of the samples in the forms of H_2_O and CO_2_, causing the loss of free hydroxyl groups at the surface of magnesium oxide, whereas the Mg atom is solidified [21].

The accumulation of impurities on the surface of magnesium oxide nanoparticles could be the reason they have a larger specific surface area, as shown in Table 2. When calcined at a higher temperature, the mesoporous structure collapses and further decreases the specific surface area [46]. The increment in the pore size distribution provided high quantities of active adsorption sites and facilitated fast mass transfer for the arsenate adsorption. Adding on, the improvement of magnesium oxide nanoparticles from a smoother to a rougher structure could aid the retention of arsenate species by holding them onto the pores, which could not be achieved through the smooth surface of the uncalcined sample.

As shown in Figure 3, the nitrogen adsorption-desorption isotherm of calcined particles was used to obtain information about the BET-specific surface area and pore size distribution. Note that the drop in the vertical axis in the BJH curve should be attributed to the fact that the amount of N_2_ adsorbed by the particles decreased with decreasing pressure. The nitrogen adsorption-desorption isotherm exhibited a type III curve according to the IUPAC classification. The BJH pore size distribution is broad in the range of 2.99–131.17 nm, indicating the presence of mesopores and macropores [47]. This is in accordance with the increase in pore size distribution as stated in Table 2.

#### 3.1.2. Surface Functionality

The X-ray diffraction pattern for both uncalcined and calcined spectra confirms that the samples are periclase, the only crystalline form of magnesium oxide. All of the reflection peaks in Figure 4 for both spectra can be indexed to the pure cubic phase of magnesium oxide. The average crystallite size, *D*, of both samples was calculated from the diffraction peaks using the Debye-Scherrer equation, as in Equation (11):(11)$$D=\frac{K\lambda }{\beta \cos\theta }$$
where *β* is the full width at half maximum height (FWHM) of the diffraction peak at an angle (in Radians), *λ* is the wavelength in (nm) of the XRD, which is 0.15418, and *K* is a dimensionless shape factor, with a value normally taken as 0.94 [48]. A slight increment in the average particle size happened from 28.40 to 34.68 nm after calcination as shown in Table 3. This agrees with the SEM micrographs obtained in Figure 1, whereby defined and clearer grains were obtained after calcination. The increased grain structures provided multiple binding sites for arsenate ions to get adsorbed easily.

The main functional groups on the surface of uncalcined and calcined magnesium oxide nanoparticles before and after adsorption of arsenate were identified using FTIR. Initially, before calcining the nanoparticles, they were tested to identify the occurrence of adsorption. As depicted in Figure 5a, no obvious bands were observed as both FTIR spectra before and after adsorption remained the same. After calcining, the original magnesium oxide particles seem to have exhibited minor changes. The comparison can be made through the different band gaps formed at different wavelengths by the respective samples, as shown in Figure 5b. The FTIR spectrum has shown characteristic bands at 3697.02 and 3648.81 cm^−1^, which are attributed to the H-O-H stretching vibrations of water molecules [49,50], while 1581.41 cm^−1^ in the same spectrum happened due to their bending vibrations [51]. The band 1396.269 cm^−1^ indicated the asymmetrical and symmetrical stretching vibrations of carboxylate (O–C=O), which could be due to the presence of minor impurities in the precursors used during the synthesis process [52]. In some circumstances, this band might also indicate the presence of aromatic rings [53]. The spectrum of 773.35 cm^−1^ basically implies the absorption band of Mg-O [53]. The bands associated with H-O-H vibrations in calcined particles completely disappear due to the decomposition of water molecules, and they are revealed back at the wavenumber of 3692.69 cm^−1^ on the As(V) adsorbed spectrum due to the bonding of As-OH in water when adsorption happened. The bands at 2359.58 cm^−1^ and 2341.74 cm^−1^ are due to the hydrocarbon chain (C-H), which is highly affected by the precursor chosen for the preparation of commercial MgO [54,55]. In the calcined particles’ spectrum, the C=O stretching vibrations were obtained around 1254.52 cm^−1^, 1118.08 cm^−1^, and 823.97 cm^−1^ [56]. The 782.51 cm^−1^ in the As(V) adsorbed magnesium oxide corresponds to As-O-Mg vibrations [57,58]. A broad band around 433–769 cm^−1^ is usually assigned to metal-oxygen bonding [59]. Therefore, in this study, Mg-O-Mg vibrations could be the reason for the formation of bands in all the spectra. As shown in Figure 5c, from 500–700 cm^−1^, the number of bands formed in the calcined particles’ spectrum is greater than the other two spectra. The calcination process has decomposed the oxides in the uncalcined nanoparticles and hence activated the Mg components, which could also be proven by the elemental composition results from Figure 2. However, after removal of As(V) ions, the peak intensities were somewhat decreased, as shown in Figure 5b, and the number of bands formed at this range was reduced too, which indicates the involvement of these functional groups in the adsorption process [60].

### 3.2. Adsorption Kinetics

From Figure 6, it was found that the adsorption capacity of magnesium oxide calcined at 650 °C for arsenate increased as a function of time in the first 195 min. The arsenate adsorption took place at a high rate in the first 75 min, then leveled off to equilibrium after 195 min. The fast adsorption at the initial stages is due to the abundant active sites on the pores of the calcined adsorbent, which become saturated with time.

Table 4 lists out the experimental and calculated adsorption capacities as well as the kinetic parameters obtained from pseudo-first order and pseudo-second order for three different initial concentrations. From the values of the correlation coefficient (*R*^2^) of both pseudo-first-order and pseudo-second-order kinetic models obtained from the non-linear plot of arsenate adsorbed by the calcined nanoparticles versus time (Figure 6), it was observed that the experimental data well matched the pseudo-first-order kinetic models. This is due to the correlation regression values of the pseudo-first-order model (*R*^2^ = 0.9819–0.9896), which are much larger compared to the pseudo-second-order model (*R*^2^ = 0.9724–0.9850). Furthermore, the gap between experimental and calculated values of $Q_{e}$ for the first-order model is narrower compared to the other model. This suggests that the arsenate adsorption by calcined magnesium oxide particles obeys a pseudo-first-order kinetic model and that the reaction is more prone to physisorption [61]. Normally, physisorption is driven by electrostatic force between oppositely charged adsorbates and adsorbents, but for chemisorption, even charged ions can interact by chemical bond [62]. In this case, physisorption can be predicted due to the electrostatically attracted Mg^2+^ ions in the activated magnesium oxide nanoparticles and the negatively charged arsenate ions. There are also physisorption-related studies involving adsorbents that show that surface charge is significantly controlled by both the point of zero charge (PZC) of the adsorbents and the heavy metal species [63]. Therefore, the dependency of the adsorption of arsenate by the calcined magnesium oxide nanoparticles on the PZC of the adsorbent might be another reason for this reaction to follow the first-order model. The rate constants of the pseudo-first-order model, $K_{1}$ increased as the concentration of arsenate solution increased. At the optimum concentration of 50 ppm of arsenate solution, the respective $K_{1}$ value decreased. At a higher concentration of 50 ppm, the calcined nanoparticles are fully saturated with the attachment of negatively charged arsenate ions, causing a lack of active sites for more adsorption to take place. The accumulation of negative charges on the surface of the adsorbents causes repulsion and further decreases the pseudo-first-order rate constant [64].

Elovich equation describes chemisorption mechanisms in nature. In order to study that, the kinetics data were fitted using the Elovich model using Equation (12):(12)$$Q_{t}=\frac{1}{\beta }ln\left(1+\alpha \beta t\right)$$
where *α* is the initial rate of adsorption and *β* is the desorption rate constant. According to the Elovich curve plotted in Figure 4, the Elovich parameters extracted are listed in Table 5.

In order to investigate whether arsenate adsorption by the calcined magnesium oxide nanoparticles was controlled by intraparticle diffusion or not, the Weber-Morris intraparticle diffusion model was implemented to fix the kinetic experimental data. The equation shown is as shown in Equation (13):(13)$$Q_{t}=K_{w}t^{0.5}+C_{w}$$
where $K_{w}$ is the rate constant of the Weber-Morris intraparticle diffusion model and $C_{w}$ represents the boundary layer effects of the adsorption. According to the equation, the plot of $Q_{t}$ versus $t^{0.5}$ should be a straight line with a slope $K_{w}\;$ and y-intercept $C_{w}$. The film diffusion is negligible, and the intra-particle diffusion is considered to be the rate-controlling step when this linear plot passes through the origin. Whereas, when the straight line does not pass through the origin (C with a value), this indicates that there is a difference in the rates of mass transfer in the initial and final steps of adsorption and that film diffusion is involved simultaneously with intra-particle diffusion. Both can be considered rate-controlling steps [65]. Based on the graph of $Q_{t}$ versus $t^{0.5}$ as shown in Figure 7 and Table 5, the graph of arsenate adsorption by the calcined magnesium oxide nanoparticles was bi-linear, suggesting that this kind of adsorption could involve intra-particle diffusion by having $R^{2}$ in the range of 0.95067–0.98669. However, the lower values of correlation coefficients, $R^{2}\;$ (0.67073–0.92785) for the whole adsorption process indicated intraparticle diffusion was not the rate-limiting step for three different concentrations of arsenate solutions by the nanoparticles. Larger values of $C_{w}\;$ also indicated that the Weber-Morris plot does not cross the origin, which strengthens the theory that intraparticle diffusion is not implied in arsenate adsorption. On the other side, the *R*^2^ values obtained from the Elovich model (0.9295–0.9758) seem to be more favorable, supporting the fact that the adsorption is slightly inclined towards chemisorption. The *R*^2^ value increases with increasing concentration, denoting the formation of increased chemical linkages. However, the *R*^2^ values for pseudo-first order are much higher than the *R*^2^ values for the Elovich model. Therefore, the occurrence of chemisorption could be there, but the abundant pore spaces that enable the diffusion of adsorbate into the adsorbent seem to be more prevalent than the chemical interaction taking place during the adsorption process. The higher *R*^2^ values in Figure 7 at the initial stage of adsorption are due to the rapid attachment of arsenate ions to the porous structure of the calcined particles, which happened to be deeper than the uncalcined ones. As the process proceeds further, the pores become occupied with arsenate, not allowing intraparticle diffusion. However, the electrostatic interaction of Mg atoms with negatively charged arsenate species and the formation of hydrogen bonds between water molecules caused a complexation reaction to take place, supporting the mild occurrence of chemisorption shown by the Elovich model.

### 3.3. Adsorption Isotherm

Furthermore, in this study, over the examined arsenate concentration range, the experimental results presented in Figure 8 were fitted to the Langmuir (Equation (6)) and Freundlich (Equation (7)) isotherm models. The best fit of the experimental data was obtained using the model with a larger $R^{2}$ value. The Langmuir and Freundlich adsorption isotherm models were applied to explore the surface properties and mechanism of adsorption of arsenate by calcined magnesium oxide nanoparticles.

From Table 6, with the higher regression coefficient, $R^{2}$, the adsorption of arsenate by the calcined magnesium oxide nanoparticles occurs through the Freundlich model with the $R^{2}$ value of 0.9980, which shows heterogeneous and multilayer adsorption on the surface of the adsorbent. The heterogenous behavior of this adsorption refers to the non-uniform adsorption sites on the adsorbent surface area. Multilayer formation has been observed when molecules are adsorbed through weak forces (long-range forces, normally under physical adsorption) due to cohesive forces exerted by the molecules of the adsorbate. The $K_{f}$ and *n* are Freundlich constants, which correspond to adsorption capacity and adsorption intensity, respectively. The *n* value indicates the degree of nonlinearity between solution concentration and adsorption as follows: if *n* = 1, then adsorption is linear; if *n* < 1, then adsorption is a chemical process; if *n* > 1, then adsorption is a physical process. The *n* value in the Freundlich equation was found to be 5.062, as shown in Table 6, indicating physisorption and supporting the assumption made in the previous kinetic isotherm study. The situation *n* > 1 is most common and may be due to a distribution of surface sites or any factor that causes a decrease in adsorbent-adsorbate interaction with increasing surface density. The values of *n* within the range of 1–10 show that the adsorption is favorable [66,67]. The higher $K_{F\;}$ value of 112.78 L/g through this study shows that the temperature for calcination used for modifying magnesium oxide nanoparticles is very suitable for arsenate adsorption compared to any other arsenate adsorption studies in previous years [68,69]. The essential features of the Langmuir isotherm can be expressed in terms of a dimensionless constant called the separation factor ($R_{L}$ also called the equilibrium parameter), which is defined by the following equation:(14)$$R_{L}=\frac{1}{1+K_{L}C_{O}}$$
where $C_{O}$ (mg/L) is the initial arsenate concentration and $K_{L}$ (L/mg) is the Langmuir constant related to the energy of adsorption. The value of $R_{L}$ indicates the shape of the isotherms to be either unfavorable ($R_{L}$ > 1), linear ($R_{L}$ = 1), favorable (0 < $R_{L}$ < 1), or irreversible ($R_{L}$ = 0). The $R_{L}$ values listed in Table 7 for all the concentrations reveal a stronger and more favorable adsorption process for this study, as the $R_{L}$ value is closer to value 0 with increasing concentration.

### 3.4. Effects of PH

As shown in Figure 9, the highest adsorption capacity for this study was achieved at a pH of 10. The trend in the adsorption capacity of the adsorbents in binding with the arsenate showed a spike at a pH of more than 8. This is in agreement with the zeta potential results obtained before and after adsorption of As(V) ions by the calcined particles at the range of pH (2–12) in Figure 10. This is because the adsorbent has a PZC of 8.7. The positively charged surface above the PZC value aided the adsorption process through electrostatic attraction. As shown in Figure 10, the calcined magnesium oxide nanoparticles become negatively charged after the adsorption of arsenate ions due to the capturing effects of arsenate.

Figure 11 shows the changes in the adsorption capacity when the dosage of 0.5 g/L is increased further. With further increases in the dosage of calcined magnesium oxide nanoparticles, the adsorption performance has degraded due to the saturation of the active sites.

Figure 12 shows the regeneration efficiency of arsenate using fresh and recycled adsorbent that has been treated with NaOH at a 1.0 M concentration. NaOH solution was selected as a regenerant solvent by many researchers for heavy metal adsorbents due to electrostatic repulsion between negatively charged arsenic species and the negative surface charge that develops on the nano-adsorbent at pH values above their PZC values [70]. By treating the used adsorbents with a 1.0 M NaOH solution, the regeneration efficiency for the second cycle is 64.01%. As expected, the regeneration efficiency of arsenate decreased with the increasing number of cycles due to the exhaustion of the adsorbent. Since physisorption appears to be the dominant adsorption process in this study, it can be interpreted that the physical changes that have taken place after adsorption on the adsorbent’s surface might be the reason for this reduction in regeneration.

## 4. Conclusions

The batch adsorption process of pentavalent arsenate using calcined magnesium oxide nanoparticles at 650 °C is dominated by physisorption at the initial stage, followed by chemisorption at the end of the process. The rapid diffusion of arsenate ions into the porous nanoparticles causes the structure to get saturated, followed by a complexation reaction between adsorbate and adsorbent. Furthermore, the abundant pore formation and grain structures formed through calcining the nanoparticles facilitate multilayer adsorption, as proven through the Freundlich isotherm model. This process is also pH-dependent, as it is solely controlled by the PZC of the adsorbent. There is no intraparticle diffusion involved in this study. Further increasing the adsorbent dosage from 0.5 g/L is not suggested for batch adsorption processes, as this is the optimum amount favorable for As(V) adsorption. The adsorption mechanism for the removal of arsenate ions from water occurred through electrostatic interactions, complexation reactions (hydrogen bonds), and porosity. The calcined magnesium oxide nanoparticles possessed decent recyclability when used up to two times. The decrease in regenerative capability is because of the physical transformation of the nanoparticles, where they become worn out with increasing adsorption. This study focuses on obtaining a high adsorptive capacity for arsenate even with a lower adsorbent dosage. The calcined magnesium oxide prepared in this study is able to adsorb arsenate through both physisorption and chemisorption but is dominated by physisorption due to its nature as a porous material. Therefore, applying this adsorbent in the form of adsorptive membranes could still be possible and could contribute to the regeneration issue because the physical structure is less likely to get damaged in the membrane matrix. Hence, with proper loading of adsorbents in membranes, calcined magnesium oxide nanoparticles at a temperature of 650 °C could be able to provide double-way adsorption for arsenate ions.

## Figures and Tables

**Figure 1 membranes-13-00475-f001:**
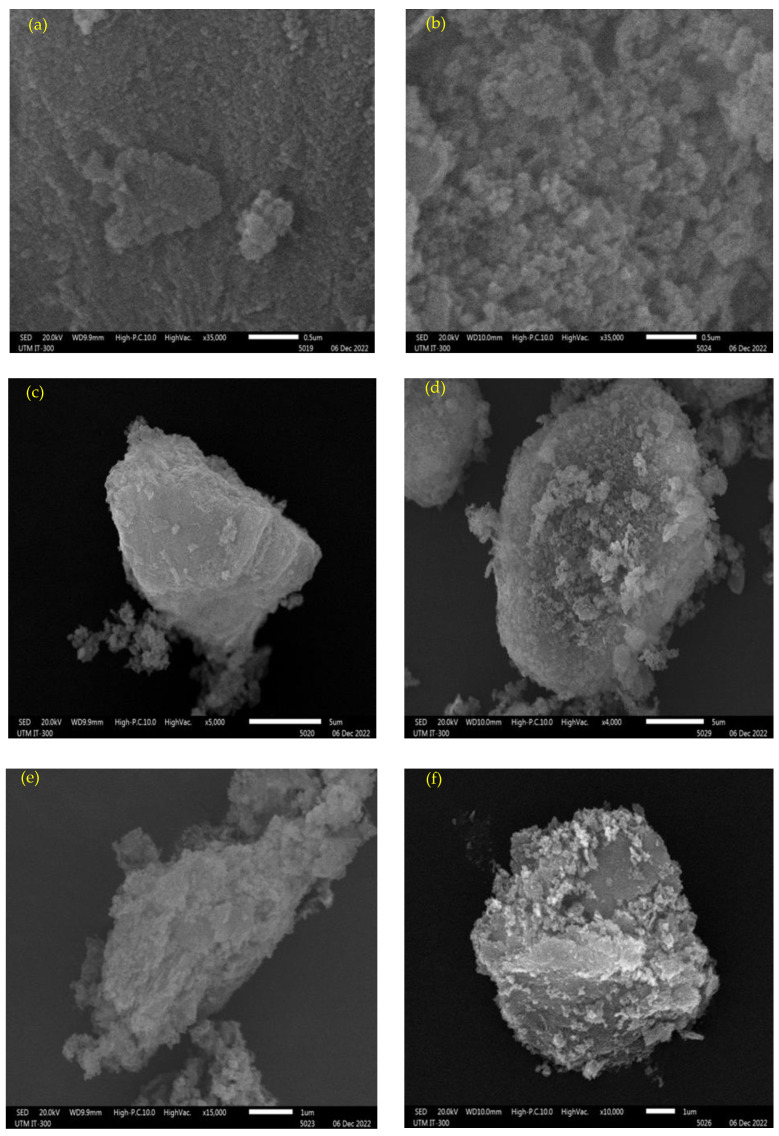
SEM micrographs of uncalcined (**a**,**c**,**e**) and calcined (**b**,**d**,**f**) magnesium oxide nanoparticles.

**Figure 2 membranes-13-00475-f002:**
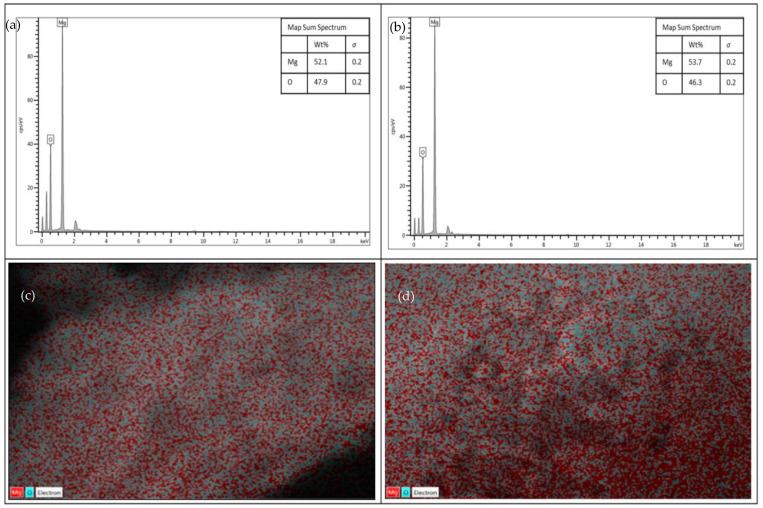
Composition of Mg and O in uncalcined (**a**) and calcined (**b**) magnesium oxide nanoparticles and their respective EDX mapping images, (**c**) uncalcined and (**d**) calcined.

**Figure 3 membranes-13-00475-f003:**
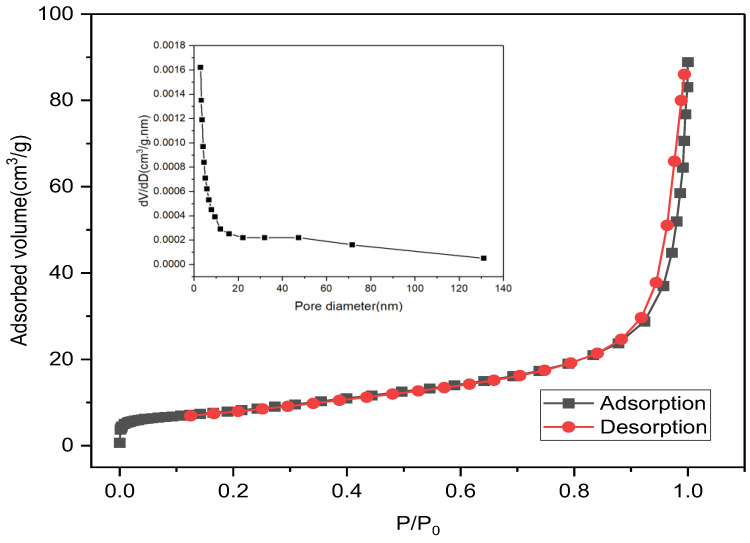
BET surface area analysis of calcined magnesium oxide nanoparticles: adsorption-desorption isotherms.

**Figure 4 membranes-13-00475-f004:**
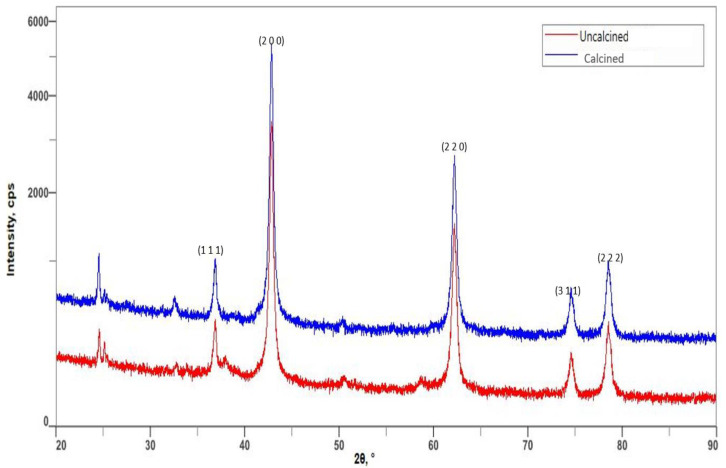
XRD spectra of uncalcined and calcined magnesium oxide particles.

**Figure 5 membranes-13-00475-f005:**
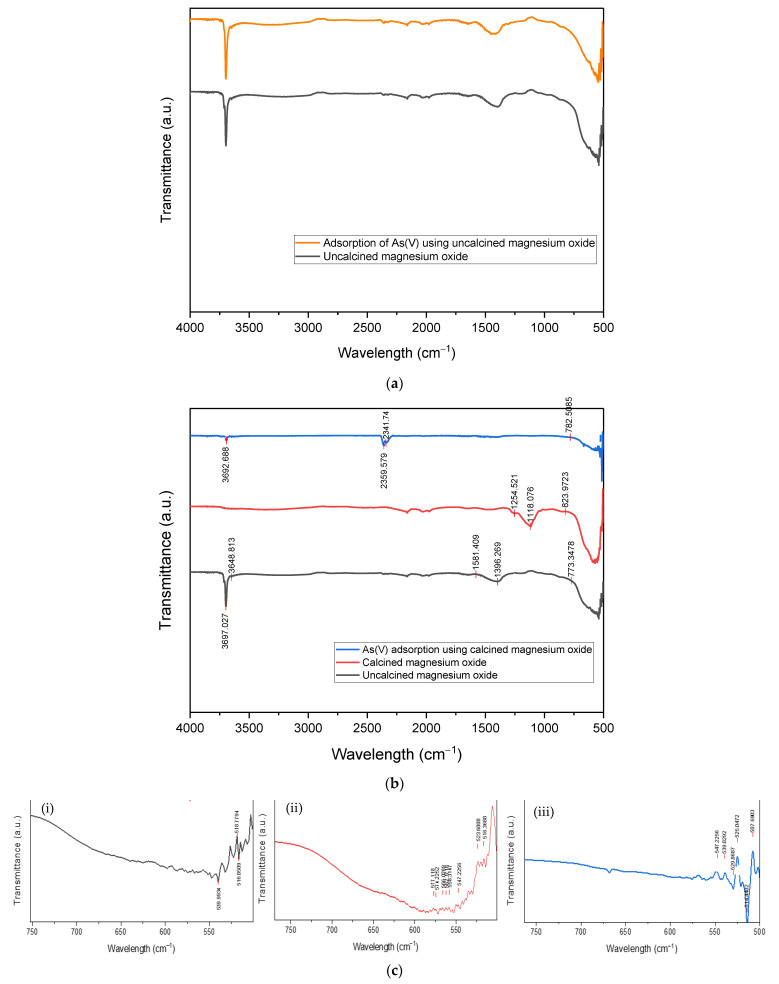
(**a**) FTIR spectrum of uncalcined magnesium oxide before and after adsorption of arsenate ions; (**b**) FTIR spectra of uncalcined, calcined, and arsenate-adsorbent magnesium oxide nanoparticles; (**c**) The Mg-O vibration bands of three spectra: (**i**) uncalcined, (**ii**) calcined, (**iii**) As(V) adsorbed.

**Figure 6 membranes-13-00475-f006:**
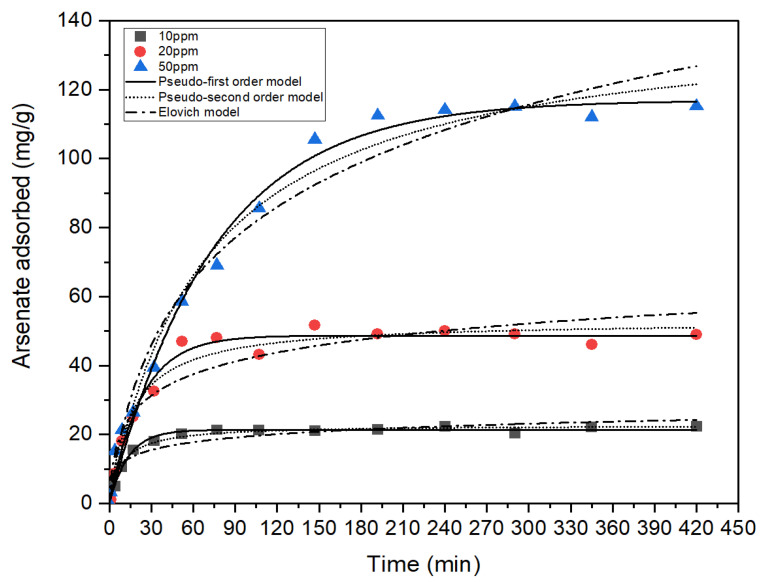
Adsorption kinetic modeling of arsenate adsorption on calcined magnesium oxide nanoparticles at different initial concentrations of arsenate solution (pH = 7.0 ± 0.1, adsorbent dosage = 0.5 g/L, contact time = 7 h).

**Figure 7 membranes-13-00475-f007:**
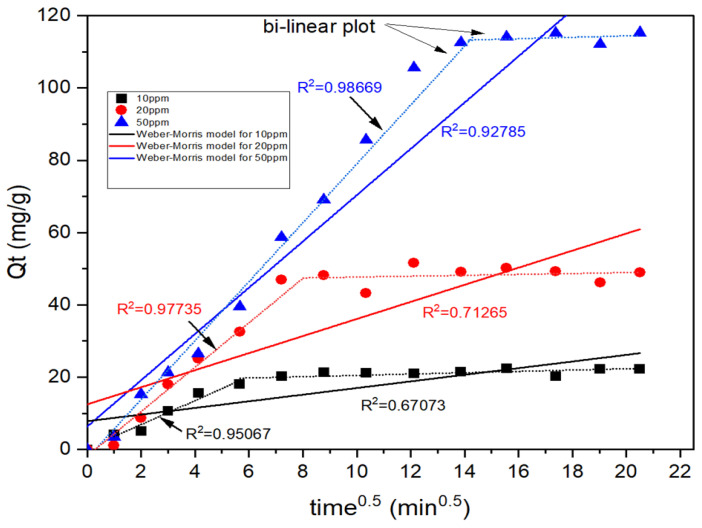
Weber-Morris intraparticle diffusion model for arsenate adsorption by calcined magnesium oxide nanoparticles.

**Figure 8 membranes-13-00475-f008:**
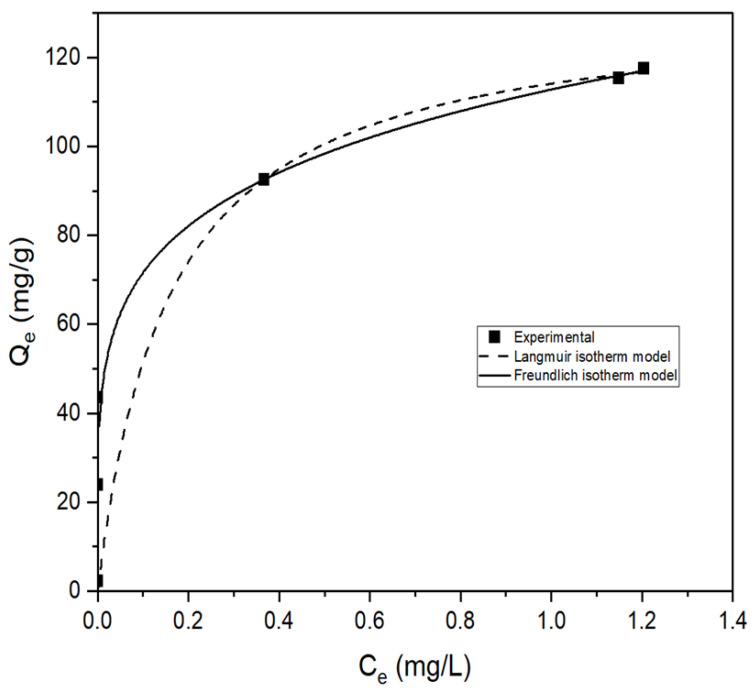
Adsorption isotherm modeling of arsenate adsorption on calcined magnesium oxide nanoparticles by Langmuir and Freundlich (pH = 7.0 ± 0.1, adsorbent dosage = 0.5 g/L, contact time = 7 h).

**Figure 9 membranes-13-00475-f009:**
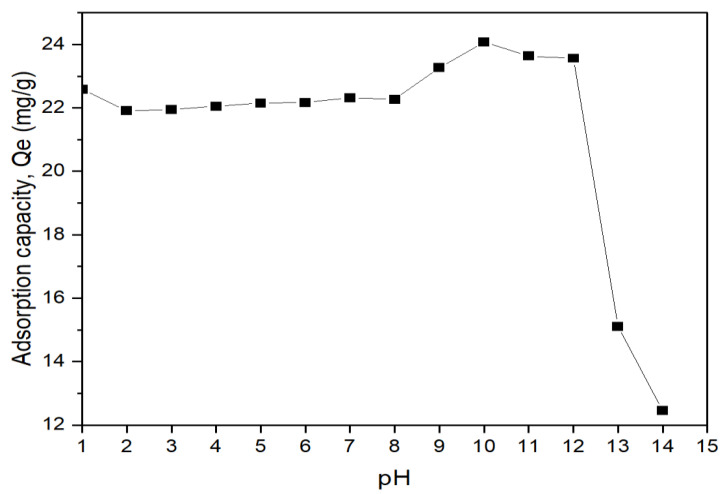
Effect of pH on arsenate adsorption by calcined magnesium oxide nanoparticles (concentration = 10 ppm, adsorbent dosage = 0.5 g/L, contact time = 7 h).

**Figure 10 membranes-13-00475-f010:**
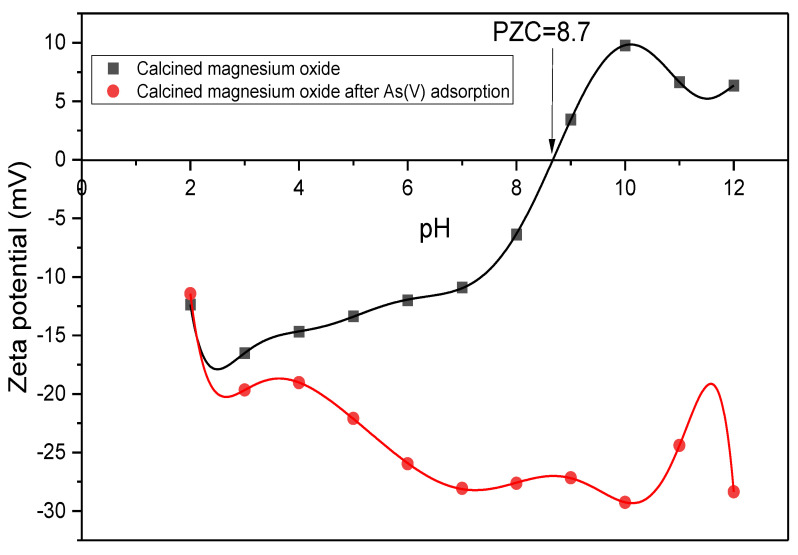
Surface charge of calcined magnesium oxide nanoparticles before and after adsorption.

**Figure 11 membranes-13-00475-f011:**
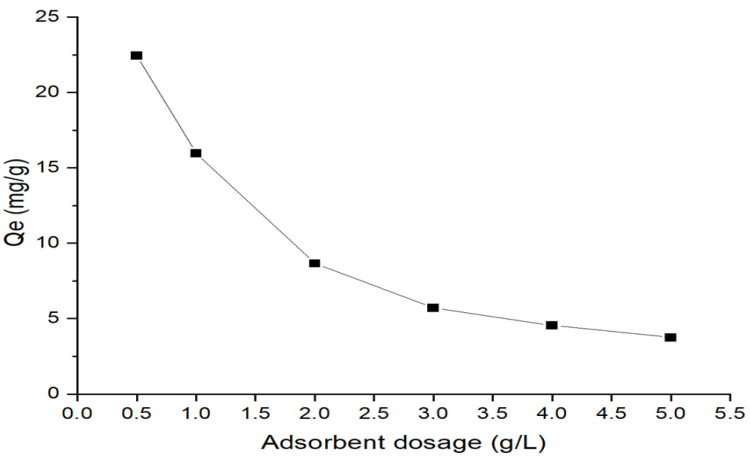
Effect of adsorbent dosage on arsenate adsorption by calcined magnesium oxide nanoparticles (concentration = 10 ppm, contact time = 7 h).

**Figure 12 membranes-13-00475-f012:**
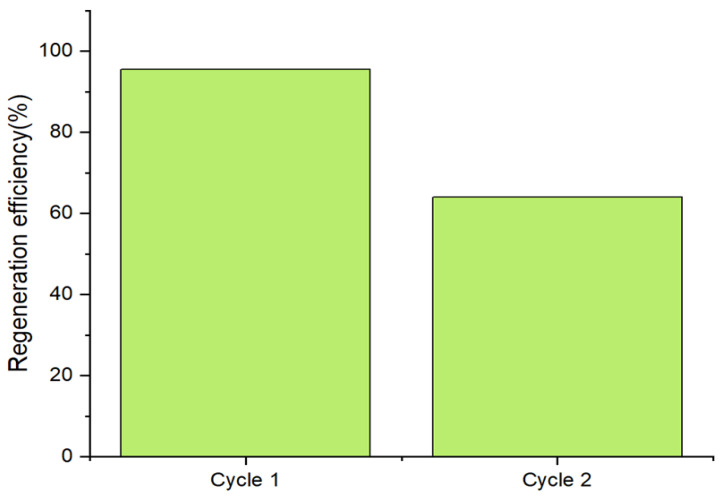
Regeneration analysis of calcined magnesium oxide adsorbents for two cycles.

**Table 1 membranes-13-00475-t001:** Comparison of arsenate adsorption studies using magnesium-associated adsorbents.

Adsorbent	Experimental Conditions	$Q_{max}$ (mg/g)	Reference
Magnesium-aluminium anchored on magnetic biochars	pH 5, T = 10 °C	34.45	[38]
Polymorphous magnesium oxide nanoparticles	-	101	[20]
Mg–Al-layered double hydroxides-functionalized hydro-char composite (single component system)	-	56.299	[39]
Mg–Al-layered double hydroxides-functionalized hydro-char composite (binary-component system)	-	16.222	[39]
Mg/Al/Fe-CLDH	pH = 2–11	70.7	[40]
Mg-doped magnetite nanoparticles by recycling of titanium slag	pH = 8–11, T = 25 °C, concentration = 10 mg/L	33.71	[41]
MgO nanoplates through the vacuum calcination route	Concentration = 40 mg/L	481	[42]
Magnesium functionalized highly ordered mesoporous Fe/Mg_4_-MCM-41 (magnesium accounts for 4%)	Adsorbent dosage = 0.5 g/L, concentration= 10–60 mg/L, pH = 3	71.53	[43]
MgO-650 °C	pH = 7, concentration = 50 mg/L, Contact time = 7 h, adsorbent dosage = 0.5 g/L	$Q_{experimental}$ = 115.27 $Q_{max}$ = 131.93	This study

**Table 2 membranes-13-00475-t002:** Physical properties of uncalcined and calcined magnesium oxide nanoparticles through BET analysis.

Sample	$S_{BET}$ (m^2^g^−1^)	PSD (nm)	$V_{SP}$ (cm^3^g^−1^)	Phase
Uncalcined	72.02	39.323	0.0717	Mesoporous
Calcined	55.72	45.907	0.0535	Mesoporous

$S_{BET}$ = BET specific surface area obtained from nitrogen adsorption data in the P/P_0_ range from 0.1 to 0.6; $V_{SP}$ = single-point pore volume calculated from the adsorption isotherm at P/P_0_ = 0.95; PSD = pore size distribution determined by using the BJH method from the adsorption branch.

**Table 3 membranes-13-00475-t003:** XRD peaks of both uncalcined MgO and MgO-650 and the respective particle sizes calculated.

MgO-650 °C			Uncalcined MgO		
2*θ* (°)	FWHM (°)	D (nm)	2*θ* (°)	FWHM (°)	D (nm)
24.51	0.15	56.99	24.56	0.16	53.52
32.54	0.40	21.71	25.11	0.19	45.57
36.85	0.28	31.11	32.81	0.32	27.00
42.85	0.32	28.17	36.82	0.22	39.18
50.37	0.18	52.20	37.84	0.97	9.05
62.22	0.36	27.24	42.85	0.31	28.49
74.58	0.33	31.47	50.50	0.73	12.62
78.53	0.38	28.53	58.70	0.47	20.27

**Table 4 membranes-13-00475-t004:** Kinetic parameters of pseudo-first-order and pseudo-second-order kinetic models for arsenate adsorption by magnesium oxide calcined at 650 °C.

First-Order Model				
$C_{o}$	$Q_{e}$ **exp**	$Q_{e1}$	$K_{1}$	$R^{2}$
mg/L	mg/g	mg/g	min^−1^	-
10	22.32	21.39	0.07	0.9819
20	48.99	48.71	0.04	0.9831
50	115.27	116.96	0.01	0.9896
**Second-Order Model**				
$C_{o}$	$Q_{e}$ **exp**	$Q_{e2}$	$K_{2}$	$R^{2}$
mg/L	mg/g	mg/g	min^−1^	-
10	22.32	22.79	0.0048	0.9818
20	48.99	53.00	0.0012	0.9724
50	115.27	141.38	1.0359	0.9850

**Table 5 membranes-13-00475-t005:** Intraparticle diffusion parameters obtained from the Weber-Morris model and the Elovich model parameters for arsenate adsorption by calcined magnesium oxide nanoparticles.

Experimental		Weber-Morris Model			Elovich Model		
$C_{o}$	$Q_{e}exp$	$C_{w}$	$K_{w}$	$R^{2}$	* **α** *	* **β** *	$R^{2}$
mg/L	mg/g	-	mg/g·min	-	g/mg·h	g/mg	-
**10**	22.32	7.806	0.917	0.6707	11.667	0.3006	0.9295
**20**	48.99	12.456	2.363	0.7127	8.728	0.1082	0.9221
**50**	115.27	6.293	6.402	0.9279	3.214	0.0291	0.9758

**Table 6 membranes-13-00475-t006:** Isotherm parameters of the Langmuir and Freundlich models for arsenate adsorption by calcined magnesium oxide nanoparticles.

Langmuir	$Q_{m}\;\;($ mg/g)	131.93
	$K_{L}\;$ (L/mg)	6.40
	$R^{2}$	0.8545
Freundlich	*n* (dimensionless)	5.062
	$K_{F}$ (L/g)	112.78
	$R^{2}$	0.9980

**Table 7 membranes-13-00475-t007:** Separation factor ($R_{L})$ of adsorption of arsenate by calcined magnesium oxide nanoparticles.

Concentration (mg/L)	1	10	20	30	40	50
* **R_L_** *	0.135	0.015	0.008	0.005	0.004	0.003

## Data Availability

The data that support the findings of this study are available from the corresponding author upon reasonable request.

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
