# Peer review of "Magnesium Oxide Nanoparticles for the Adsorption of Pentavalent Arsenic from Water: Effects of Calcination"

_membranes, 2023, doi:10.3390/membranes13050475_

Round 1

Reviewer 1 Report (Previous Reviewer 2)

Corrections have been made by the authors. A manuscript may applicable for publication.

Author Response

Reviewer 2 Report (New Reviewer)

In this work, the authors synthesized magnesium oxide particles at high temperatures and used these particles for the adsorption of arsenic ions in aqueous solutions. The authors observed the material using various techniques such as BET, SEM, and EDX, and claimed that the pore structure helped the material achieve adsorption of arsenic ions. Considering that water treatment is currently an important research direction, valuable research is worth publishing. However, before publication, I believe the following issues are worth discussing.

Regardless of the reason, it is evident that such magnesium oxide particles are not reusable, which means that this type of material is unlikely to be adopted on a large scale. So, what is the practical significance of this research?

I find it difficult to see the pore structure that the authors claim based on the SEM images. Is the fit of the authors' BET data reasonable? Are there any other methods to characterize the pore structure?

From the SEM images, the morphology and size of these particles appear highly non-uniform, with some larger particles exceeding 15 micrometers. It is worth questioning whether such a material should be referred to as 'nanoparticles'.

The scale bar in the SEM images and the axis of Figure 2 are too blurry.

Author Response

Reviewer 3 Report (New Reviewer)

The subject of this study (arsenic adsorption by using calcined MgO nanoparticles) could fit this journal, but additional explanation on the relationship between the subject of this work and membrane technology should be provided. Additinally, several flaws to be revised were found. The publication of this manuscript should be reconsidered after the substantial revisions listed below.

(1) In the Introduction, the authors described that the influence of calcination conditions (calcination atmosphere and temperature) have been scarcely discussed (line 121-124), but MgO was calcined at one temperature (650ºC) in air in this study. Thus the effects of calcination environments and temperatures have not been elucidated. This inconsistency between the aim and the results should be corrected.

(2) 3.1.1., Fig. 1, Table 2: The calcined MgO exhibits fracture formation on their surface (Fig.1) and  the accumulation of impurities was associated with the imcrease in specific surface area (line 299-300). However, the S(BET) of the calcined MgO was smaller than that of uncalcined MgO. These results seem to be inconsistent and should be checked again.

(3) Resolution of some figures is quite low. For example, magnifications of Fig. 1 are hard to be read.

(4) line 343: The IR absorption bands found at 2359 and 2341 cm-1 have been assigned to hydrocarbon C-H vibrations. What are the origins of these bands? (Are any hydrocarbons contained in the sample?)

(5) line 456-464, 467-473: These parts look redundant and contain mainly common knowledges. It is better to reconstruct these parts.

(6) line 364: As(IV) ---> As(V)?

(7) Usage of capital/lowercase letters should be carefully checked again. For example, is it really necessary to spell the name of MgO always using capital letters (Magnesium Oxide)?

Round 2

Reviewer 2 Report (New Reviewer)

The authors have made some modifications to the paper based on my feedback to some extent. However, I still hold my position that if the material is not reusable, I do not believe it has research value in water treatment. From this perspective, I find it difficult to support its publication. Nevertheless, the authors have conducted quite detailed research in other aspects. This work may provide some valuable information for the field of water treatment.

Author Response

Reviewer 3 Report (New Reviewer)

This manuscript has been satisfactorily revised.

Some comments for further improvement:

(1) Response 3: Uncalcined magnesium oxide has a higher specific surface area compare to calcined one. This is due to the deposition of impurities on magnesium oxide before calcination and they disappear slowly with calcination.

--->  Taking account of the removal of impurities through calcination and resultant decrease in specific surface area, the calcined MgO surface is expected to be smoother than that before calcination. However, the calcined MgO surface looks rougher rather than the surface of uncalcined MgO (Fig. 1(b), (d)  and (f), line 331-337).  The changes in surface roughness and specific surface area seem to be inconsistent and may cause confusion. More consistent explamation could be necessary.

(2) Resolution is Fig. 2 is also low. For example, labels and numbers of the axes are hard to be read.

(3) Response 8: The term “MgO” has been changed to magnesium oxide. The author has modified the name of adsorbents as “calcined and uncalcined magnesium oxide nanoparticles.” 

---> Usage of chemical formula (MgO) is no problem. When the name of MgO is spelled, spelling with lowercase letters (magnesium oxide) could be more common than using capital letters (Magnesium Oxide).

(4) Please check the number of digits appeared in the main text. For example, details numbers such as 782.5085 cm-1 (FTIR) and 2.9932 nm (BJH pore size) are found; are all of them really so accurate?

Round 3

Reviewer 2 Report (New Reviewer)

I have no further comments

This manuscript is a resubmission of an earlier submission. The following is a list of the peer review reports and author responses from that submission.

Round 1

Reviewer 1 Report

The manuscript presented by Mehanathan et al. presented the As(V) adsorption of “MgO nanoparticles”. However, there is insufficient data to prove all hypothetical assumptions, the results indicate that nanoparticles of pure MgO were obtained by calcination. However, not XRD or TEM measurements were carried out to prove the presence of pure MgO. SEM and FTIR are complementary characterization techniques. Since particle size determination is key to understand the adsorption phenomena, which has a dependence on the adsorption comprehension by this adsorbent, the manuscript cannot be accepted in Membranes in the present form. We cannot speculate on the adsorption outcomes without knowing the size and, of course, the BET area parameters. Furthermore, no comparison with other extensive literature on As(V) adsorption is identified and collected in a Table. As a result, the significance of this study is not unique/novel in the field of heavy metal adsorption/removal.

Reviewer 2 Report

1. Why was the effect of pH studied in the range of 2 to 12? Why hasn't the wider pH range of 1 to 14 been explored?

2. The Elovich adsorption kinetic model is not presented. However, given the data obtained, it is of interest.

3. Figures 2(i) and 2(ii) are essential for evaluating the structure of the materials under study. However, they are too small and faded. The authors need to improve their display, increase the inscriptions, and most importantly, make the percentage of Mg and O in the studied materials clearer and larger.

4. The article shows only the theoretical maximum capacity obtained from the Langmuir isotherm. However, for some reason, the authors did not do enough experiments to establish a practical maximum capacity. I consider it a must.

5. The conclusions need to be made more detailed.